# Variational-LSTM autoencoder to forecast the spread of coronavirus across the globe

**Mohamed R. Ibrahim**[1]*, **James Haworth**[1], **Aldo Lipani**[1], **Nilufer Aslam**[1], **Tao Cheng**[1], **Nicola Christie**[2]

1 SpaceTimeLab, Department of Civil, Environmental and Geomatic Engineering, University College London (UCL), London, United Kingdom, 2 Department of Civil, Environmental and Geomatic Engineering, Centre for Transport Studies (CTS), University College London (UCL), London, United Kingdom

* mohamed.ibrahim.17@ucl.ac.uk, mibrahim2006@me.com

## Abstract

Modelling the spread of coronavirus globally while learning trends at global and country levels remains crucial for tackling the pandemic. We introduce a novel variational-LSTM Autoencoder model to predict the spread of coronavirus for each country across the globe. This deep Spatio-temporal model does not only rely on historical data of the virus spread but also includes factors related to urban characteristics represented in locational and demographic data (such as population density, urban population, and fertility rate), an index that represents the governmental measures and response amid toward mitigating the outbreak (includes 13 measures such as: 1) school closing, 2) workplace closing, 3) cancelling public events, 4) close public transport, 5) public information campaigns, 6) restrictions on internal movements, 7) international travel controls, 8) fiscal measures, 9) monetary measures, 10) emergency investment in health care, 11) investment in vaccines, 12) virus testing framework, and 13) contact tracing). In addition, the introduced method learns to generate a graph to adjust the spatial dependences among different countries while forecasting the spread. We trained two models for short and long-term forecasts. *The first one* is trained to output one step in future with three previous timestamps of all features across the globe, whereas *the second model* is trained to output 10 steps in future. Overall, the trained models show high validation for forecasting the spread for each country for short and long-term forecasts, which makes the introduce method a useful tool to assist decision and policymaking for the different corners of the globe.

## 1. Introduction

As a novel contagious disease, COVID-19 has reached more than eight millions confirmed cases and more than 400,000 death globally by 14[th] of June 2020 [1]. Although there are a number of the statistical and epidemic models to analyse COVID-19 outbreak, the models are suffering from many assumptions to evaluate the impact of intervention plans which create a low accuracy as well as unsure prediction [2]. Therefore, there is a vital need to develop new frameworks/methods to curb/control the spread of Coronavirus immediately [2, 3].

**Data Availability Statement:** The raw data used for conducting this research can be found in these links (as accessed by 15/06/2020): 1) Covid-19 spatio-temporal data – GitHub Repository: https://github.com/CSSEGISandData/COVID-19 2) Urban

features https://www.worldometers.info/world-population/population-by-country/ 3) Governmental responses and Stringency index: https://github.com/OxCGRT/covid-policy-tracker We also added a zip folders, "datasets_raw_files", containing these data files respectively. 1) time_series_19-covid-Confirmed.csv 2) population.csv 3) OxCGRT_latest.csv

**Funding:** The authors received no specific funding for this work.

**Competing interests:** The authors have declared that no competing interests exist.

The epidemic outbreak of COVID-19 in literature is investigated using mathematical compartmental model named Susceptible-Infected-Recovered (SIR) [4]. The SIR model represents a population under three categories: 1) Susceptible (the number of people presently not infected), 2) the number of people currently infected, and 3) the number of people either recovered or died. The model describes as differential equations. The model is completely determined by transmission rate, the recovery rate, and the initial condition, which can be estimated using least square error, Kalman filtering or BMC. The model is sometimes renamed based on the new parameters such as Susceptible-Infectious-Quarantined-Recovered (SIQR) or Susceptible-Exposed-Infected-Recovered (SEIR). The main idea in the version of all SIRs models are four-fold; first, identification and better understanding current epidemic [5], second, simulation the behaviour of the system [6], third, forecasting of the future behaviour [7], and last, how we control the current situation [8]. However, the results of the models including accuracy only valid based on their assumptions in a slice of available data/moment and have their scopes to assist healthcare strategies for the decision-making process.

On the other hand, agent-based modelling is utilised to explore and estimate the number of contagions of COVID-19, specifically for certain countries [9, 10]. Also, statistical methods [11], simple time series modelling [12], and logistic map [13] are utilised for similar objectives, whereas [3], focused on modelling the spread of coronavirus based on the parameters of basic SIR in a (3-dimensional) iterative maps to provide a wider picture of the globe. Petropoulos and Makridakis [14] forecasted the total global spread relying on exponential smoothing model based only on historical data. Put all together, the drawbacks of their models are not flexible to fit for each country or region due to the lack of necessary measures, government responses, and spatial factors related to each specific location.

There are few examples of predictive modelling of the coronavirus spread based on machine learning approaches, whether through shallow or deep models. While it is can be explained due to the limitation of data since the early stage of the outbreak, it remains an essential tool. According to Pham and Luengo-oroz [15], machine learning approaches certainly could assist in forecasting by with improved quality for prediction. One of the few studies is presented by [2]. They have applied real-time short-term forecasting using the compiled data from 11[th] Jan to 27[th] Feb 2020 collected by the World Health Organization (WHO) for the 31 provinces of China. The data is trained on a deep learning model for real-time forecasting of new cases for the provinces. Their model has the flexibility to be trained at the city, provincial, or national level. Besides, the latent variable of the trained model is used to extract necessary features for each region and fed into a K-means to cluster similar features of the infected or recovered features of patients. Bearing this in mind, there is still a knowledge gap for machine learning models to predict coronavirus cases at a global as well as regional scales [15].

While SIR models with their different types, in addition to the aforementioned ones, are essential, the challenges remain in forecasting different regions and countries across the globe with a single model without any assumptions or scenario-based rules, but only with the current situations, features related to countries, and measures amid to reduce the impact of the outbreak. Accordingly, in this paper, we introduce a new method of learning and encoding information related to the historical data of coronavirus per country, features of countries, spatial dependencies among the different countries, and last, the time and location-dependent measures taken by each country amid towards reducing the impact of Coronavirus. Relying on deep learning, we introduce a novel variational Long-Short Term Memory (LSTM) autoencoder model to forecast the spread of coronavirus per country across the globe. This single deep model aimed to provide robust assistance to policymakers to understand the future of the pandemic at both a global level and country level, for a short-term forecast and long-term one.

The main advantages of the proposed method are: 1) It can structure and learns from different data sources, either that belongs to spatial adjacency, urban and population factors, or various historical related data, 2) the model is flexible to apply to different scales, in which currently, it can provide prediction at global and country scales, however, it can be also applied to city level. And last 3) the model is capable of learning global trends for countries that have either similar measures, spread patterns, or urban and population features.

After the introduction, the article is structured in five sections. Section 2 introduces the method and materials used. In section 3, we show model evaluations and the experimental results at country and global levels. In section 4 we discuss our results, compare our model to any existing base models and highlights limitations. Last, in section 5 we conclude and present our recommendation for future works.

## 2. Methods

### 2.1 Hypothesis and assumptions

The model algorithms are constructed based on four assumptions that we assume the model needs to learn to predict the next day spread: First, the model needs to extract features regarding the historical data of coronavirus spread for a given country bearing in mind the historical values of the virus spread in the other countries simultaneously before it outputs a prediction for a given country. Second, before the model gives a predicted value for each country, it should consider the predicted values of all other countries instantaneously, similar to the first point. Third, the spatial relationship between different countries is multidimensional; it can vary based on geographical location, adjacency, accessibility, or even policies for banning accessibility. The model needs to deal with variations of time and location of the different inputted scenarios while sampling outcomes. Last, apart from the virus features, for each country, there are unique demographic and geographical features that show association to the spread of the virus that may show association with the virus, in which the learning process of the model needs to consider each time before it gives a predicted value.

The structure of the input data is key for any model to learn. Fig 1 shows the concept of the overall structure of the proposed graph of multi-dimensional data sets for forecasting the spread. It illustrates how different types of data can be linked and clustered for the model to learn the spread of a virus. This data can be seen as dynamic features related to both virus and the location with long temporal scales (i.e. the population data) or short ones ($t_i$). It shows how local and global trend for a virus can be forecasted for a given country ($n_z$), with urban features that include both spatial and demographic factors ($x_m$), that share a spatial weight ($g_j$) with other countries in the graph, whereas government mitigated measures ($r_q$) are applied. Put all together, the model needs to differentiate between factors that characterise countries or regions, and those which characterise the virus spread to understand the patterns of spread at global and country levels.

### 2.2 Translation to the machine

To meet these hypotheses and assumptions during the learning process, the architecture of the proposed model is based on the combinations of three main components: 1) LSTM, 2) Self-attention, and 3) Variational autoencoder graph.

**2.2.1 LSTM cells.** LSTM represents the main component of the proposed model. It has been shown it is the ability to learn long-term dependencies easier than a simple recurrent architecture [16, 17]. Unlike traditional recurrent units, it has an internal recurrence or a self-loop, in which it allows the timestamps to create paths, in which the gradient of the model can flow for a long duration without facing the vanishes issues presented in a normal recurrent

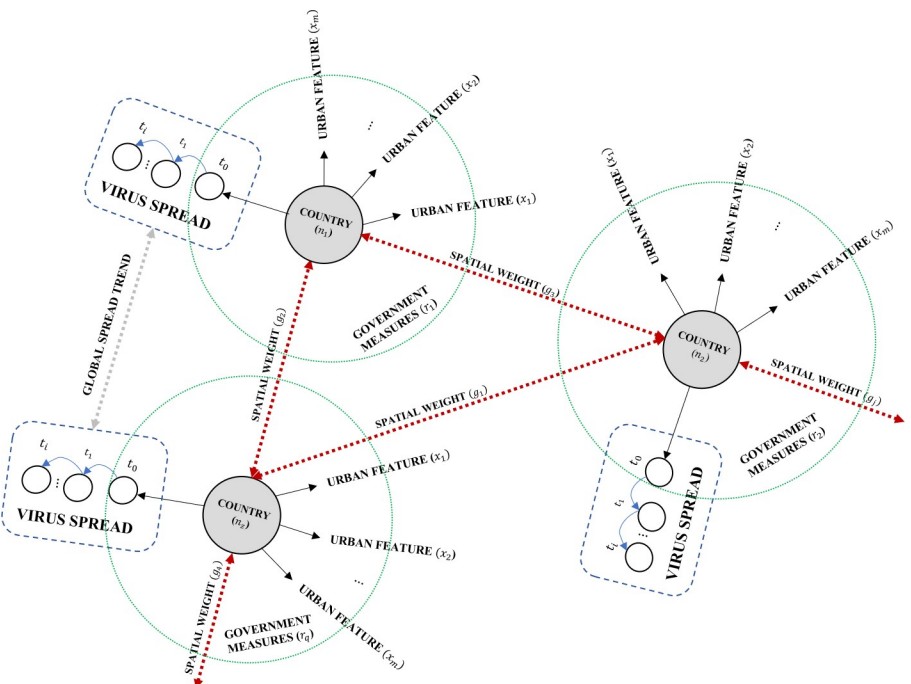

**Fig 1. The concept for structuring the graph for the proposed variational-LSTM autoencoder.**

unit. Even for an LSTM with a fixed parameter, the integrated time scale can change based on the input sequence, simply because the constants of time are outputted by the model itself. These self-loops are controlled by a forget gate unit ($f_i^{(t)}$) for a given time (t) and a cell (i), in which it fits this weight to a scaled value between 0,1 with a sigmoid unit ($\sigma$). It can be explained as:

$$f_i^{(t)} = \sigma\left(b_i^f + \sum_j U_{i,j}^f x_j^{(t)} + \sum_j W_{i,j}^f h_j^{(t-1)}\right) \quad (1)$$

Where $x^{(t)}$ is a vector for the current input, $h^{(t)}$ is a vector for the current hidden layer that contains the outputs of all the LSTM cells, $b^f$ are the biases for the forget gates, $U^f$ is the input weights, $W^f$ is the recurrent weights for the forget gates.

The internal state of the LSTM is updated with a conditioned self-loop weight ($f_i^{(t)}$) as:

$$s_i^{(t)} = f_i^{(t)} s_i^{(t-1)} + g_i^{(t)} \sigma\left(b_i + \sum_j U_{i,j} x_j^{(t)} + \sum_j W_{i,j} h_j^{(t-1)}\right) \quad (2)$$

Where b represents biases, U represents input weights, W represents the current weights into the LSTM cell, and $g_i^{(t)}$ represents the external input gate unit. It is computed similar to the forget gate but with it is own parameters as:

$$g_i^{(t)} = \sigma\left(b_i^g + \sum_j U_{i,j}^g x_j^{(t)} + \sum_j W_{i,j}^g h_j^{(t-1)}\right) \quad (3)$$

Last, the LSTM cell output $h_i^{(t)}$ can also be controlled and shut off with an output gate $q_i^{(t)}$,

similar to the aforementioned gate by using a sigmoid unit. The output $h_i^{(t)}$ is computed as:

$$h_i^{(t)} = _\varphi(s_i^{(t)})q_i^{(t)} \tag{4}$$

$$q_i^{(t)} = \sigma\left(b_i^o + \sum_j U_{i,j}^o x_j^{(t)} + \sum_j W_{i,j}^o h_j^{(t-1)}\right) \tag{5}$$

Where $b^o$ represents biases, $U^o$ represents input weights, $W^o$ represents the current, and $_\varphi.$ represents the activation function such as tanh function.

Put all together, this controls of the time scale and the forgetting behaviour of different units allow the model to learn long- and short-term dependencies for a given vector. Not only the model learns from the previously defined timestamps for each country, but also the model could extract features from the other countries at each given timestamp in which the dimension of the input vector, and cell states, includes the dimensions of the different countries. It is worth mentioning that the input for the LSTM cells is can be seen as a three-dimensional tensor, representing the sample size for both training and testing, the defined timestamps for the model to look back, and the timestamps of the other countries as a global feature extractor.

**2.2.2 Self-attention mechanism.** While the LSTM cells learn from their input sequence to output the predicted sequences through the long and short dependencies of the time constants and their additional features for each country, the relations between its inputs remains missing. A self-attention mechanism allows the LSTM units to understand the representation of its inputs by relating the positioning of each sequence [16, 18]. This mechanism in the case of the proposed model is crucial to assist the model to which piece of information to consider and what to forget when making a prediction.

**2.2.3 Variational autoencoder graph.** We initialise the first graph based on the spatial weight of the geographical locations of all infected countries (more details will follow in subsection 3.1.4), however, despite the attempts of trying to create a sophisticated adjacency matrix among the infected countries (based on flight routes, spatial network, migration network, etc.), the output may misleading for any learning method over time or for a given location. The spatial weight since the outbreak of the model may look completely different from the initial day to the latest day. These due to different policies and measures that are taken by countries. However, due to its high uncertainty and variation. Inputting the model with a static graph or even a dynamic one based on limited data may exacerbate the learning process. Accordingly. the third vital components in our model represent the variational autoencoder (VAE) component that allows the model to generate information from a given input. It can be defined as a generative directed method that makes use of the learned approximate inference [16, 19]. The model is based on the idea of passing latent variables $z$ to the coded distribution $p_{model}(z)$ over samples $x$ using a differentiable generator network $g(z)$. Subsequently, $x$ is sampled from the distribution of $p_{model}(x; g(z))$ which is equal to the distribution of $p_{model}(x|z)$. The model is trained by maximising the lower bound of the variation $\mathcal{L}(q)$ that belongs to $x$ as:

$$\mathcal{L}(q) = \mathbb{E}_{z \sim q(z|x)} \log p_{model}(z, x) + \mathcal{H}(q(z|x)) \tag{6}$$

Eq (6) describes the joint log-likelihood of the visible and hidden variables under the approximate posterior over the latent variables $\log p_{model}(z,x)$, and the entropy of the approximate posterior $\mathcal{H}(q(z|x)$, in which $q$ is chosen to be a Gaussian distribution with a noise that is added to the predicted mean value. In traditional VAE, the reconstruction log-likelihood tries to equalise the approximate posterior distribution $q(z|x)$ and the model prior $p_{model}(x|z)$. However, in the case of our model the encoded $q(z|x)$ is conditioned and penalized based on

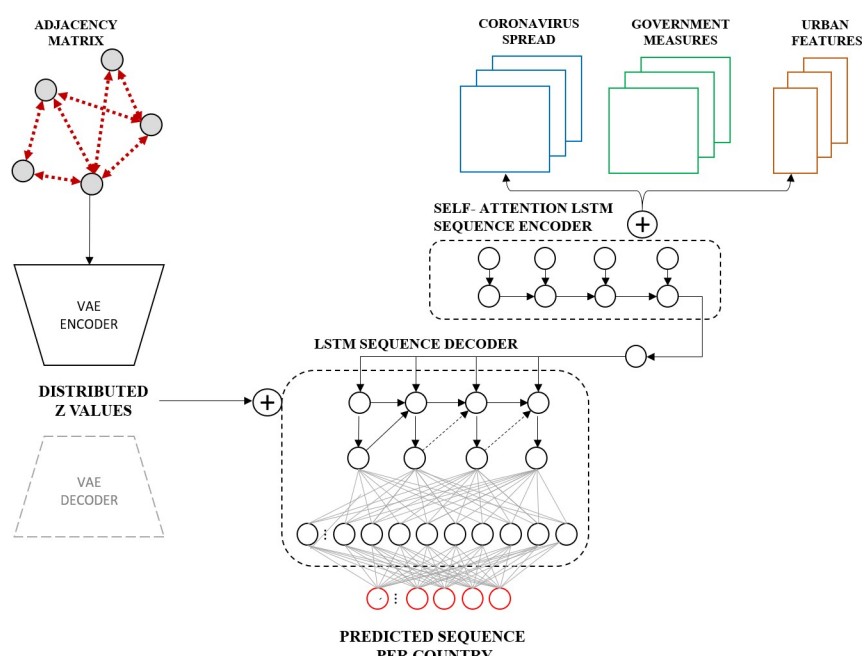

**Fig 2. The proposed variational LSTM autoencoder model.**

the output of a predicted value of the next forecast of the spread, instead of the log-likelihood of the similarity with $p_{model}(x|z)$, which will be explained further in the proposed framework.

## 2.3 Proposed model framework

We propose a sequence-to-sequence architecture relying on a mixture of VAE and LSTM. The model comprises two branches trained in parallel in an end-to-end fashion. Fig 2 shows the overall proposed framework.

The first branch is a self-attention LSTM model that feeds by Spatio-temporal data of coronavirus spread per day and per country, the government policies per day and per country, and the urban features per country, in which the vector is repeated to cover the duration of training (The urban features used are three features: population density, urban population percentage and fertility rate, which will be covered in detail in the upcoming section). Each input is reshaped as a 3D tensor of shape (samples, timestamps, number of features X number of countries). The three-input data are concatenated at the last axis of the data (the dimension of the feature) and passed to the first branch of the model through two parts: 1) the self-attention LSTM sequence encoder, and 2) the LSTM sequence decoder.

The first sequence encodes the input data and extracts features for the second part of the LSTM sequence to output the prediction of the spread for the next day (in case of the short-term forecast) per country.

The first part consists of three LSTM layers, each consists of 50 LSTM units. The first two layers activated by a Rectified Linear Unit (ReLU) and a separated by a Dropout layer of size (0.2) to minimise over-fitness Moreover, a self-attention mechanism is applied after the second LSTM layer. The final LSTM unit is activated by a linear function as the first output for the LSTM sequence encoder part.

In parallel to the self-attention encoder sequence, the second branch of the model is an encoder of VAE. It is feed by a spatial matrix of dimensions (number of countries X number

of countries) and repeated for the entire duration of training and timestamps (In the next section, more details will follow on how it is selected and computed). This encoder part is mainly a convolution structure, which consists of three 1D convolution layers of filters 32, 64, and 128 respectively, in which they are all of a kernel size of 1 and activated by a ReLU function and followed by a Dropout layer of size 0.2. After the dropout, two LSTM layers are followed, in which they contain 100, 494 LSTM cells respectively. The first one is activated by a ReLU function, whereas the second one by a linear function. A fully connected layer of neurons equivalent to the number of countries is applied. Last the latent space is defined with a dimension of 10, in which the z-values are generated from sampling over the Gaussian distribution of the previous inputted layer (As explained in section 2.2.3). To visualise the generated graph for representation purposes, It is worth mentioning that the encoder of the second branch of the model can be decoded to output the generated samples for each predicted sequence by passing it into a decoder VAE, where the 1D convolutions layers are transposed to a final output shape equal to the inputted dimension. As for future work, this could be an interesting approach to understanding the variation of the graph for each predicted day for all countries.

Both outputs of the self-attention LSTM encoder and the encoder of the VAE are concatenated over the feature dimension and passed to the LSTM decoder sequence, which contains a single LSTM layer of cell numbers equal to the total number of countries. It is followed by two fully connected layers of shape size (1 X number of countries) for predicting the value of the next day, in case of the short-term forecast, or can be shaped to (numbers of future steps X number of countries) for any number of future steps that model needs to output per each country.

Data sets are split to training and testing on the first dimension of data shape (the total duration of the temporal data), in a way that the model can be tested on the last 6 days. We trained two different models, one as a single-step model for the short-term forecast (one day), whereas the other is trained as a multi-step model (10 days forecast). There are two crucial differences between these two models; The output layer, and the dimension of the y-train, and y-test of the first one is shaped as (1 x n), whereas in the other model is output layer is shaped as (10 X n), despite the number of samples. is the structure of the y-train and y-test. The second issue is the trained and tested sample is not only reduced by the number of timestamps–at the beginning of each sequence- as in the case of the first model but also reduced by the number of future steps -at the end of the sequence- in the case of the second model. Last, based on trial and error, we structured the data based on 3 timestamps for both models to look back for all the input features for each country, in which we found optimal results.

The weights of the model are initialised by random weights. The model is compiled based on the backpropagation of error of the stochastic gradient descents, relying on 'adam' optimiser [20], with a learning rate of 0.001 and momentum 0.9. The model is trained for 500 training cycles (epochs).

## 2.4 Evaluation metrics

The performance of the proposed method is evaluated based on three different scales; 1) a global loss-based evaluation, 2) country-based evaluation and last, 3) step-based evaluation. The short-term forecast model (single-step model) relies only on the first two evaluation metrics, whereas the multi-step model includes the three levels of evaluations.

The first loss function evaluates the overall performance of the model at a global level, in which it influenced the adjustment of the model weights during training for both trained models. It is evaluated based on the Mean Squared of Error (MSE) which is calculated as:

$$MSE_{test} = \frac{1}{m} \sum_{i}^{m} \left( \hat{y}^{(test)} - y^{(test)} \right)^2 \tag{7}$$

Where m is the total sample, $\hat{y}^{(test)}$ is the predicted values of the test set, and $y^{(test)}$ is the observed values of the test set.

Furthermore, we computed a Logarithmic version of Mean squared error or so-called Mean squared logarithmic error (MSLE) to understand the ration between the true and predicted values. This function is accountable for the relative difference between the true and predicted values, whereas large errors are not significantly penalised than small ones. MSLE makes it easier for understanding and comparing the model performance in different countries despite how small or large their number of cases. MSLE is defined as:

$$MSLE = \frac{1}{m}\sum_{i}^{m}\left(log(y_i + 1) - log(\hat{y}_i + 1)\right)^2 \tag{8}$$

We also computed Kullback–Leibler divergence ($D_{KL}$) or so-called 'relative entropy 'which measures the difference between the probability distribution of two sequences. It is a common approach for assessing the VAE, nevertheless, it could be a good indicator to evaluate the predicted sequences globally. It is calculated as:

$$D_{KL}\left(p(x)\|q(x)\right) = \sum_{x\in X}p(x)\ln\frac{p(x)}{q(x)} \tag{9}$$

Where $p(x)$ and $q(x)$ represent the two probability distributions of the two random discrete sequences of $x$. In the case of the model $p(x)$ and $q(x)$ represents the true distribution of data and the predicted one ($y^{(test)}$ and $\hat{y}^{(test)}$). It is worth mentioning $\left(p(x)\|q(x)\right) \neq \left(q(x)\|p(x)\right)$.

The second loss evaluates the performance of the model at a local level of each country or region. Strictly, $\hat{y}^{(test)}$ and $y^{(test)}$ ideally fit a statistically significant linear model where the strength of the model with r-squared value can be computed for further interpretation, in addition to the computed MSE or its root, for each county for the entire duration. Similar to the second loss, the performance of the second model (the multi-step model) includes a calculated loss (based on the root of the MSE) for each predicted step.

Last, comparing our results to other models remains a challenge due to the absence of a unified model similar to what we have achieved that forecast each country globally, or also due to the absence of general benchmark data with a common evaluation metrics. However, we try our best to compare and discuss the performance of our method to any existing models such simple or deep time-series model for specific countries or at any specific time.

## 3. Materials and feature selections

### 3.1 Input data

To forecast the spread of the Coronavirus the next day, we synchronised different types of data to allow the model to learn. This wide range of data comprises the historical data of the coronavirus spread by each country, dynamic policies and government responses that amid to mitigate Coronavirus by each timestamp and by each country, static urban features that characterise each country and shows significant correlations with the virus spread, and last, the spatial weight among the different countries. These different data types are integrated and synchronised by countries and -time steps in case of dynamic data–to be feed to the introduced framework.

**3.1.1 COVID-19 confirmed cases data.** We used the historical data for Coronavirus spread published by John Hopkins University [21, 22]. After integrating this data with following data sources, the version we used, contains timestamps from 22/01/2020 till 14/06/2020 (144 days) for 282 regions or countries across the globe as shown in Fig 3 for the confirmed cases for the start and end day of the examined duration.

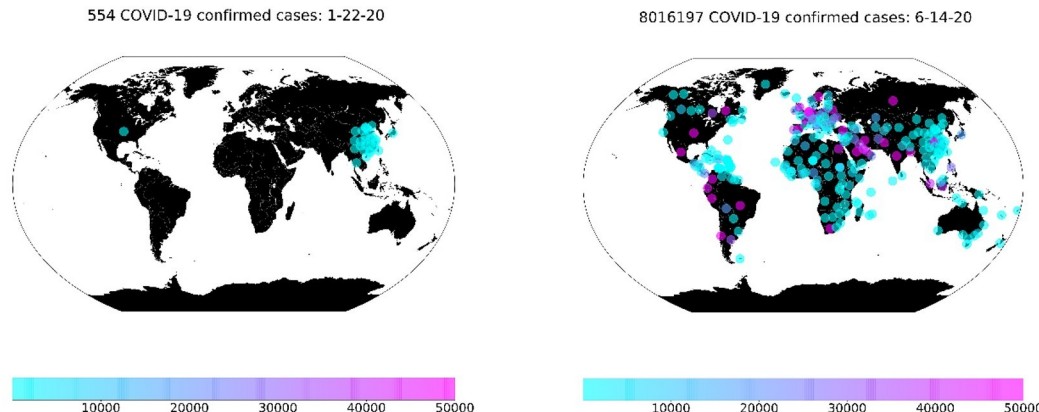

**Fig 3. Confirmed accumulated cases globally from 22/01/2020 to 14/06/2020.**

**3.1.2 Urban features data.** We used demographic and locational data that represent the population for each region or country from the aforementioned data set [23]. There is a wide range of factors, however, we only selected three factors; 1) Population density, 2) fertility rate and 3) Urban population. Fig 4 shows the spatial dynamics of these three factors. The two reasons for selecting these features are: First, the selection is based on enhancing the model prediction after several trial and errors with and without several features. Second and most importantly, the selected features show a statistically significant association with the spread of coronavirus over time for all countries across the globe. We examined other variables that represent each country or region such as the absolute value of population in 2020, the yearly change in the population, the world share of the population, and the value of the land area for a given country, in which we found them insignificant with

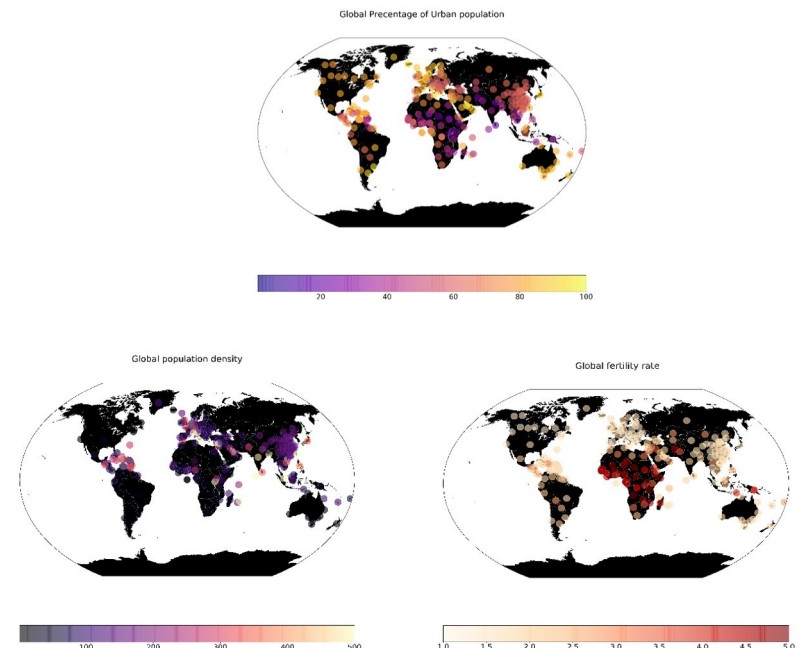

**Fig 4. The selected three factors (urban population, population density, and fertility rate).**

the spread of coronavirus over time. Fig 5 shows the outputs of the spearman correlation for the three selected factors. In Fig 5A, the population density was significant with decaying positive correlation coefficients (rho) for the first 70 days from the starting date. This means at the early stage of the virus spread, the higher the population density, the more likely a higher coronavirus spread. In Fig 5B, the fertility rates across the globe show a significant association over the entire test duration except for May. The significant results are with negative rho values, which means countries with higher fertility rates are less likely to have a higher spread of coronavirus, except for the spread of the virus in May. This could explain the less spread of the virus in Africa (as shown in Fig 3), however, this feature may be a time-dependant or due to reporting inaccuracy or the low percentage of virus testing in Africa. Last, in Fig 5C, the percentage of the urban population started to show a significant association with the spread of the virus with positive rho values only for the period between the end of February till the first week of May. During this period, this means the higher the countries with a higher percentage of the urban population, are more likely to have higher coronavirus spread.

**3.1.3 Government Response Stringency Index.** Different countries took and continuously take different measures and responses amid towards coronavirus outbreak. These time and location dependant measures include 13 indicators, which they are: 1) school closing, 2) workplace closing, 3) cancelling public events, 4) close public transport, 5) public information campaigns, 6) restrictions on internal movements, 7) international travel controls, 8) fiscal measures, 9) monetary measures, 10) emergency investment in health care, 11) investment in vaccines, 12) virus testing framework and 13) contact tracing. Put all together, Oxford COVID-19 Government Response Tracker [24] aimed to measure the variation of the government responses weighted by these indicators in a scaled index, so-called Stringency Index. It is worth mentioning that the data is continuously updated, whereas new indicators are introduced to improve the quality of the

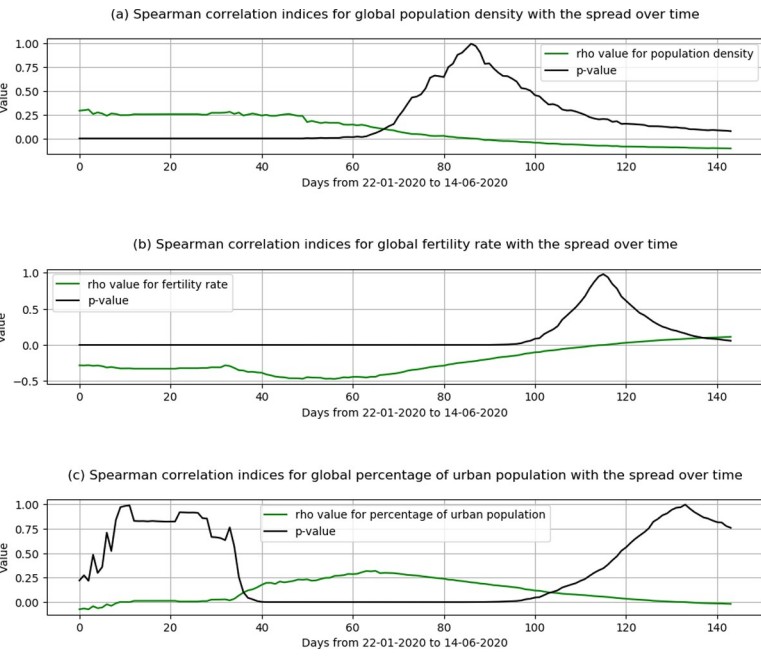

**Fig 5. Spearman correlation indices for the select urban features with the coronavirus spread (the period between 22/01/2020 to 14/06/2020).**

Stringency Index. We used this index to weight the different countries based on the government responses, after integrating and matching the time and location of the previously mentioned data sets (See Fig 6).

**3.1.4 Spatial weight.** We computed a spatially weighted adjacency matrix based on the geolocation of each region or country, relying on the geodesic distance between each region or country. We used the haversine formula to compute the distance on the sphere. It calculated as:

$$a = sin^2\left(\frac{\Delta\varphi}{2}\right) + cos\varphi_1 cos\varphi_2 sin^2\left(\frac{\Delta\lambda}{2}\right) \tag{10}$$

$$d = R\left(2 \cdot atan2\left(\sqrt{a}, \sqrt{(1-a)}\right)\right) \tag{11}$$

Where $\varphi_1$, $\varphi_2$ represent the origin and destination latitudes in radian respectively, $\Delta\lambda$ represents the change between the origin and destination longitudes in radian, and R is the earth's radius.

The adjacency matrix is conditioned based primary on eliminating long-distance connections, which can represent the connection between the US and Europe, the US and China, and direct connection between China and the rest of the world. This hypothetical assumption came from the early international measured took by the US to ban flight to Europe and China for Non-American citizens. Given, this spatial weight may vary or have a higher degree of uncertainty, the model only self-learns from its representation while it generates various samples with the VAE encoder as discussed earlier, instead of using these data as a fixed and constant factor during training and testing. to be in business-as-usual. However, these are only a few easily interpretable examples, the challenges for the model is to self-learn the representation of the graph to adjust the different weights and generate a graph that could in forecasting the spread globally.

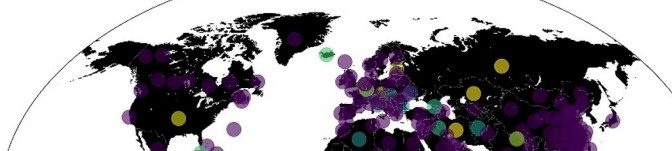

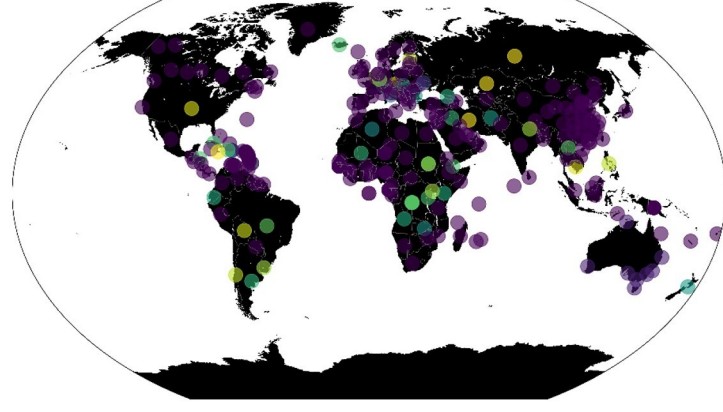

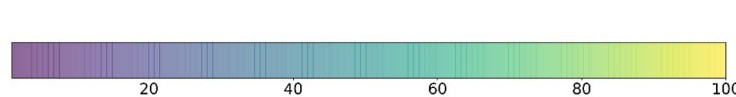

**Fig 6. Shows an example for the stringency index globally for 28/03/2020.**

In Fig 7, we show how we initialised the adjusted spatially weighted matrix for all countries. It attempts to show three main elements for computing the graph: first, it shows how a complete graph between the origin and destination countries is computed. Second, how the relative distance is computed and conditioned. And last, it shows how the array is scaled and reshaped.

Fig 8 shows examples of the variation that could be more significant and realistic for predicting a given day for a given country. For instance, the first graph in Fig 8, can represent countries with strict measures towards international travel, the second one which could be the more likely to be the case during the period of banning travel from the US to Europe or China, for instance, the last two shows how the world more likely.

## 4. Results

### 4.1 Model evaluation globally

After 500 epochs, the training and testing curves of the model show a steady output with no sign of over fitness, nevertheless, the MSE losses for both curves are at a minimum, with values less than 0.01, whereas the KL loss for the test set is less than 0.37 for both trained model. In Fig 9, we show the distribution of the confirmed and predicted cases globally with the single-step model. The total predicted cases per day is a close number to the actual data, with a slightly higher confirmed in Africa than what has been confirmed.

In Fig 10, we show the sum of the accumulated predicted cases–predicted at a country level —across the globe for each day regarding the actual data. The results are highly accurate at a

```
INPUT  number of countries (n)
INPUT  weighted_distance
INPUT  latitude
INPUT  longitude
Radius = 6372.800
FOR x  = 0 to n -1
     i = 0
     φ₁ = radian (latitude[x])
     λ₁ = radian (longitude[x])
     WHILE i < latitude
              φ₂ = radian (latitude [i]))
              λ₂ = radian (longitude [i])
              i ← i+1
     END WHILE
     a = sin²(Δφ/2)cosφ₂ * sin²(Δλ/2)
     d = Radius (2 * atan2 (√a, √(1−a) ))
      IF d < weighted_distance  THEN
         OUTPUT d
      END IF
NEXT x
END FOR
computed_distance = d → scaled [0:1]
spatial_weight = computed_distance → reshaped (n , n)
OUTPUT spatial_weight
```

**Fig 7. Algorithm 1.** Initializing the adjusted spatially-weighted adjacency matrix.

An adjusted spatially weighted adjacency matrix for all countries

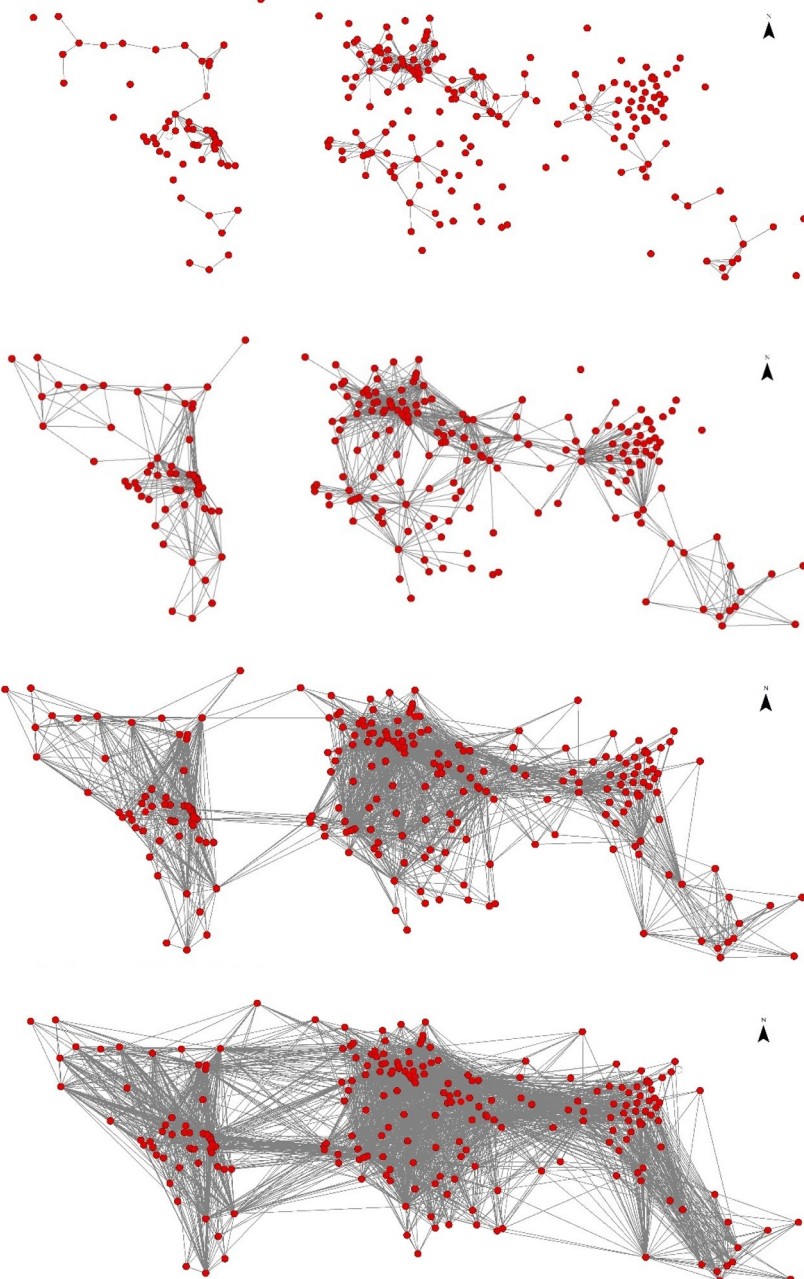

**Fig 8. Few examples of different adjusted spatially weighted adjacency matrix, conditioned by limiting direct connection that would be generated by VAE after initialisation.**

global level, with a fraction difference between the actual and predicted ones on the last examined day 14/06/2020. Specifically, Fig 10A show the accumulated prediction and the actual cases globally in a linear scale of cases counts. Fig 10B shows the true and predicted of confirmed cases at a logarithmic scale. It compares the logarithmic prediction and true values. After the initial days- where the initial cases were mainly zero in many countries globally and

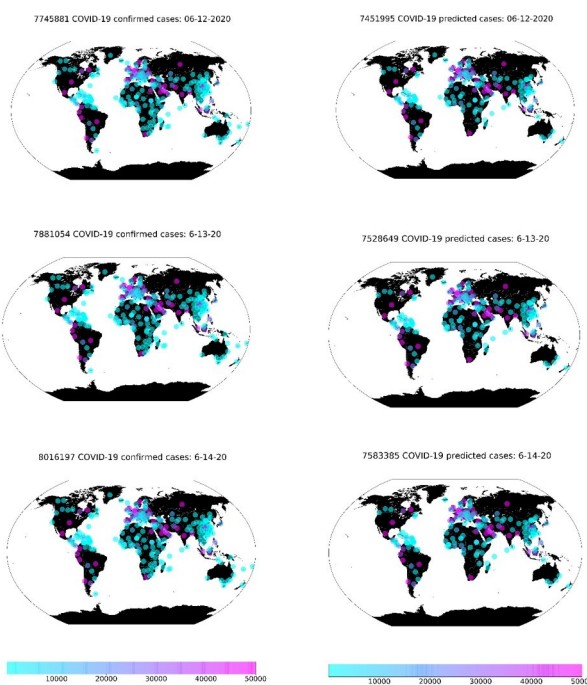

**Fig 9. Accumulated confirmed cases (left-hand side) vs predicted ones globally (right-hand side) for the last three days of the examined data (the period between 12/06/2020 to 14/06/2020).**

there was no enough data for the model to learn- the model shows a high validation in learning the overall pattern, in addition to predicting the actual numbers.

The prediction of the model is nonlinear, however, its output at a given sample when compared to its ground truth is linear. Therefore, fitting a linear regression model between the predicted result and the observed one and providing an r-squared value could be a good indicator for understanding the model strength. Fig 10C shows the relation between the confirmed and predicted cases after fitting it to a linear model. It also shows the r-squared value, the root of the MSE metrics (RMSE) and the MSLE for a linear regression fitted model on the predicted and actual values of our single-step model. The computed metrics show a high linear association among them.

What makes this method more reliable than any simple time-series model is that the predicted global curve to the actual one is outputted without the model learning any explicit rules extracted at the global level to mimic the global spread curve of the virus. The model learns the patterns at country levels, whereas error is minimised at both local and global levels. What makes this a very crucial point to discuss is that changes across the globe are more likely to happen at a country level, whereas the global level is rather an impact of the different countries.

Table 1 compares the result of the introduced method with and without the adjacency matrix and the variational component of the model. It shows that the introduced method with the variational autoencoder component and the introduced adjacency matrix enhances the prediction by reducing the model loss at a global scale with an RMSE value of 174.3 and MSLE value of 0.472. These results show the significant impact of the introduced method in understanding adjacency between different countries.

## 4.2 Evaluation of selected countries

Not only does the model shows strong performance globally but also at the country level. Fig 11 shows the performance of the single-step model in different countries (for linear

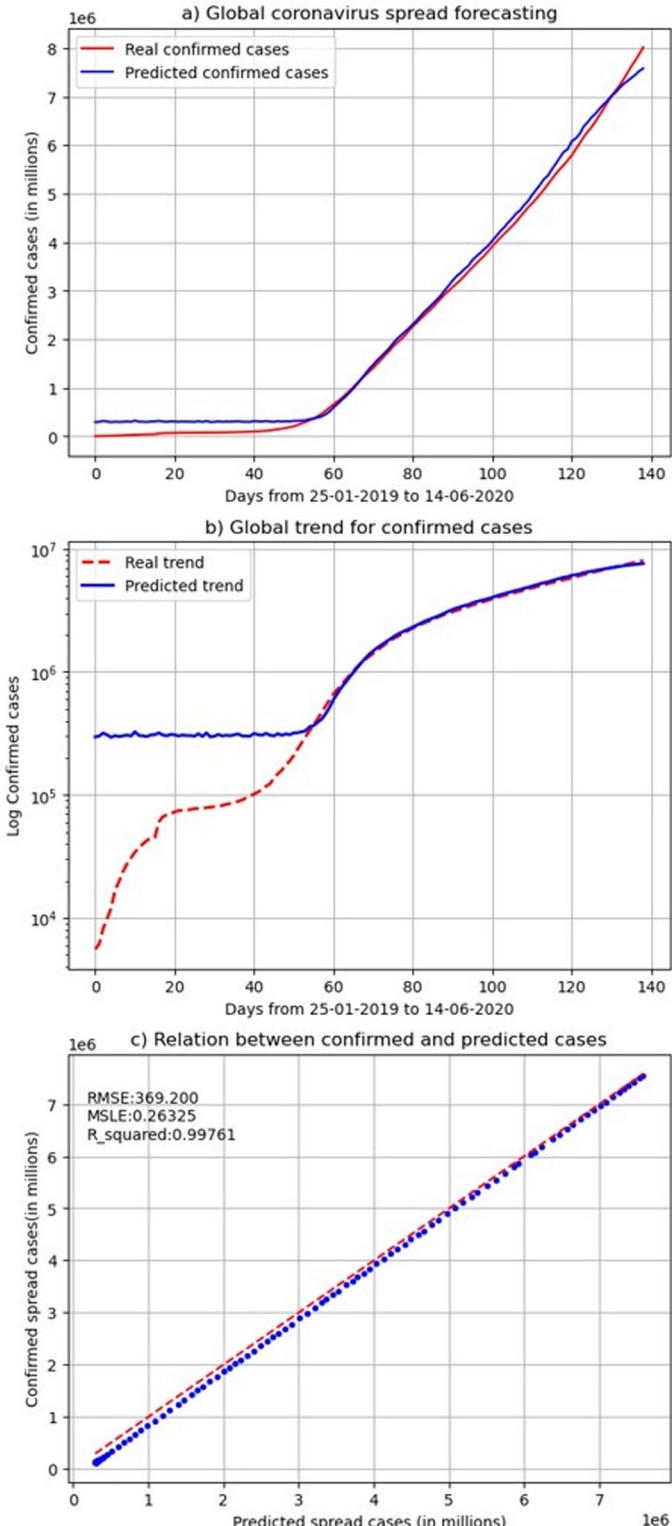

**Fig 10. The total confirmed cases globally and the sum of the predicted cases at a country level (linear and logarithmic scales).**

**Table 1. The impact of the spatially-weighted adjacency matrix on the model prediction.**

| Model losses | Our model | Method without adjacency matrix and VAE |
|---|---|---|
| Global loss (RMSE) | 369.2 | 543.5 |
| Global loss (MSLE) | 0.263 | 0.735 |

and logarithmic scales). After having enough data and after passing the initial days with zero values (the first 40–50 days), the model shows high performance in learning the spread pattern. It is worth mentioning that the distortion in ground truth curves reflects data uncertainty, which accordingly, it impacts the variance of the predicted values. Overall, the model shows higher performance in countries with higher spread whereas the performance of the model decreases with countries with fewer cases over a short period. However, the model shows overall reliable results at a country level for estimating the actual results and their overall patterns.

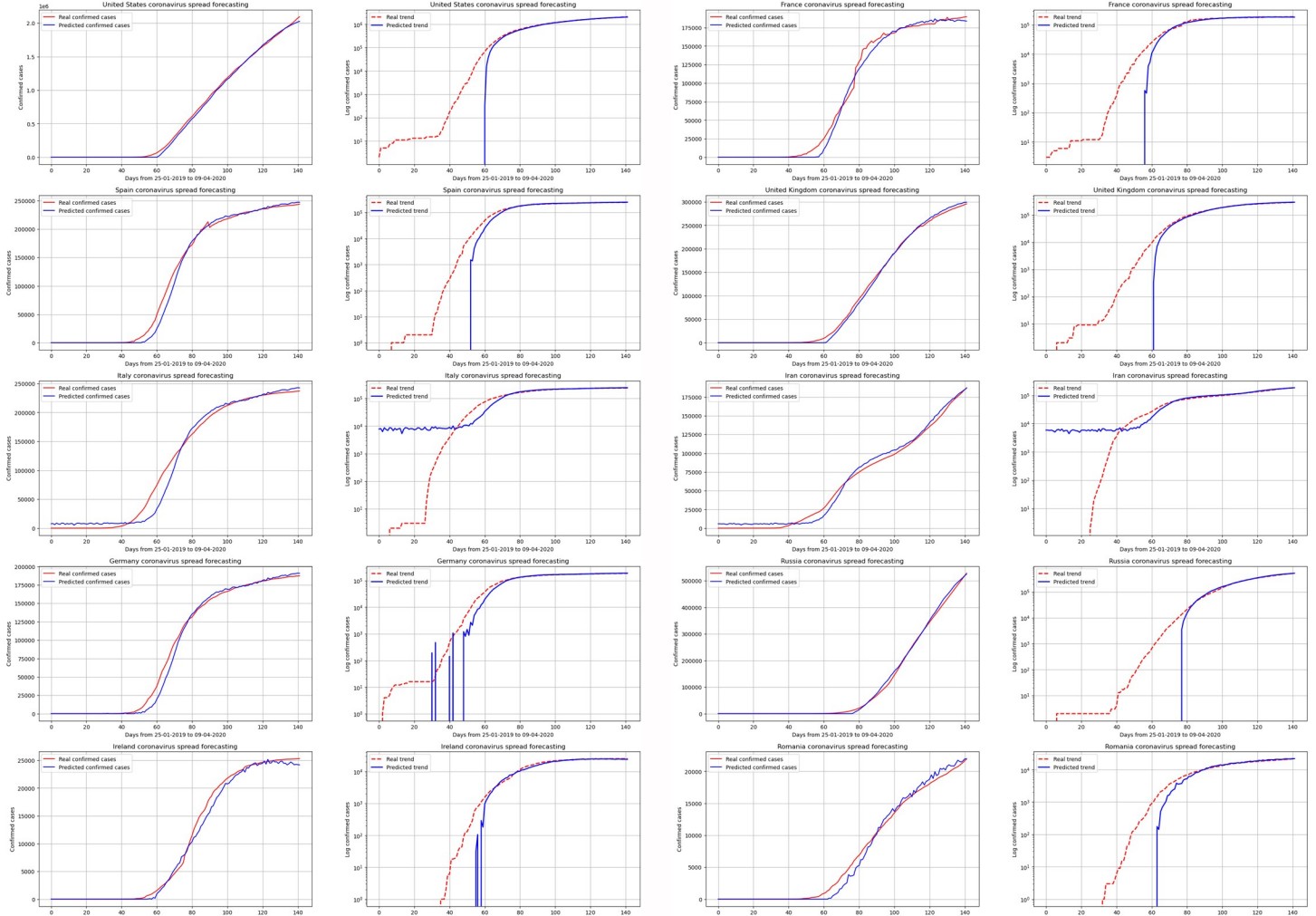

**Fig 11. Model prediction and ground truth for selected countries with three timestamps and one predicted step (the period between 25/01/2020 to 14/06/2020).**

### 4.3 Evaluation of single-step and multi-step models

In Table 2, we extend on the evaluation of the single-step model. We show a further variation of prediction in selected countries in different continents. While the model performance varies from a country to country, overall, it shows a reliable result for at a country level.

In Table 3, we show the performance of the 10-step model for a group of selected countries. This model is evaluated per country and step. While the model performance reduces with the increase of the number of steps, compared to the single-step model, the result to a higher degree remains consistent at a country level when we reach the 10-step.

## 5. Discussion

In this article, we introduce a method for predicting the spread of coronavirus for each country across the globe for both short and long-term forecast. It has three main advantages, first, the model learns not only from the historical data but also the applied governmental measures for each country, urban factors, and the spatial graph that represent the dependencies among the different countries. The second advantage of the model is its ability to be applied at various scales. Currently, it can forecast the spread at a global and country, and region level (i.e. the case of China, UK), however, it can also be applied at the city level. Last, the model can forecast short and long term forecast which could be a reliable tool for decision-making.

### 5.1 Base model evaluations

There are different attempts for relying on a simple time-series model whether it is relying on machine learning or a simple mathematical rule for a single country or the total cases globally. However, the drawback in such methods is: First, by fitting an exponential smoothing function to a model with no controlled point would mean the virus will continue to spread, regardless of the number of a population, the action is taken. Second, if a simple rule for a given country works for the last days, till when this logic will continue works? What happens when values remain constant, decrease, or even increase at a different rate? There are different possible scenarios that such an approach could not answer. Third, despite the first two arguments, how many rules are needed to fit each country globally at a longer period? Subjectively, a simple

**Table 2. Prediction evaluation of selected countries with one-step model.**

| Country/Region | RMSE | MSLE | R_squared |
|---|---|---|---|
| United States | 14130.53528 | 39.804184 | 0.999086957 |
| Spain | 4008.887015 | 36.020918 | 0.994331297 |
| Italy | 2991.006704 | 1.746805 | 0.980901649 |
| Germany | 3050.007901 | 31.859709 | 0.994348671 |
| Ireland | 482.8544471 | 47.415084 | 0.995977711 |
| France | 2610.205157 | 39.366827 | 0.994498587 |
| United Kingdom | 2684.074255 | 42.58654 | 0.998842593 |
| Iran | 2618.327467 | 3.2166192 | 0.992377009 |
| Russia | 4149.796122 | 59.220917 | 0.998912445 |
| Romania | 457.5678294 | 53.975618 | 0.993425659 |
| India | 2562.456552 | 49.584238 | 0.989854353 |
| Egypt | 985.4251867 | 57.478551 | 0.981669757 |
| Saudi Arabia | 628.0872982 | 43.871154 | 0.99574248 |
| Japan | 351.7628198 | 16.720756 | 0.98089273 |
| Sri Lanka | 790.4085479 | 97.331081 | 0.763025016 |

Table 3. Prediction evaluation of selected countries with the multi-step model (10 steps).

| Country/ Region | RMSE_1 steps | R2_1 steps | RMSE_2 steps | R2_2 steps | RMSE_3 steps | R2_3 steps | RMSE_4 steps | R2_4 steps | RMSE_5 steps | R2_5 steps | RMSE_6 steps | R2_6 steps | RMSE_7 steps | R2_7 steps | RMSE_8 steps | R2_8 steps | RMSE_9 steps | R2_9 steps | RMSE_10 steps | R2_10 steps |
|---|---|---|---|---|---|---|---|---|---|---|---|---|---|---|---|---|---|---|---|---|
| United States | 53745.07 | 0.996001 | 50758.17 | 0.997688 | 49787.91 | 0.998378 | 46380.31 | 0.99921 | 44256.98 | 0.999583 | 43014.93 | 0.999591 | 42318.31 | 0.999378 | 43565.35 | 0.998978 | 38614.66 | 0.997662 | 38650.57 | 0.996699 |
| Spain | 8135.392 | 0.951804 | 8535.279 | 0.949953 | 8080.889 | 0.944773 | 8350.676 | 0.932669 | 8589.636 | 0.933209 | 8438.724 | 0.915116 | 9087.234 | 0.898291 | 10065.5 | 0.884036 | 8776.778 | 0.879211 | 8712.039 | 0.877964 |
| Italy | 7715.221 | 0.936516 | 7540.355 | 0.922011 | 7822.895 | 0.926668 | 7148.621 | 0.917526 | 7982.064 | 0.909544 | 8384.304 | 0.898409 | 7732.904 | 0.904248 | 7780.143 | 0.898468 | 7828.845 | 0.875727 | 8059.713 | 0.867137 |
| Germany | 5904.827 | 0.963411 | 5708.251 | 0.944782 | 6249.719 | 0.940764 | 5957.042 | 0.929008 | 6248.873 | 0.932889 | 5944.218 | 0.933691 | 6115.885 | 0.910983 | 6101.513 | 0.913604 | 6146.706 | 0.890176 | 6441.707 | 0.891453 |
| Ireland | 1696.473 | 0.980609 | 579.4737 | 0.962512 | 3185.499 | 0.943999 | 1062.02 | 0.93458 | 2225.631 | 0.980171 | 1584.665 | 0.955625 | 1709.481 | 0.980283 | 2369.655 | 0.985918 | 1679.125 | 0.968656 | 1574.823 | 0.94575 |
| France | 6052.225 | 0.957675 | 6486.893 | 0.958396 | 5861.654 | 0.964344 | 6675.339 | 0.96345 | 6226.281 | 0.940845 | 6655.494 | 0.950137 | 7767.713 | 0.93135 | 6584.615 | 0.924911 | 6607.083 | 0.921949 | 6192.694 | 0.925632 |
| United Kingdom | 6676.21 | 0.993298 | 5405.892 | 0.994552 | 7036.269 | 0.994975 | 6266.128 | 0.995904 | 5153.879 | 0.995633 | 5493.48 | 0.995796 | 3877.11 | 0.99176 | 6506.995 | 0.995029 | 6225.636 | 0.994266 | 7928.694 | 0.995434 |
| Iran | 10316.2 | 0.987389 | 6541.67 | 0.980953 | 9994.938 | 0.981846 | 10878.89 | 0.978958 | 10132.63 | 0.97524 | 9772.654 | 0.972886 | 12231.1 | 0.969461 | 10635.81 | 0.965137 | 13853.68 | 0.963378 | 11765.76 | 0.959907 |
| Russia | 41736.47 | 0.929755 | 41931.45 | 0.930685 | 43600.89 | 0.922439 | 42674.37 | 0.91491 | 40678.7 | 0.935744 | 39708.76 | 0.940782 | 39884.15 | 0.94091 | 39220.35 | 0.944605 | 38220.63 | 0.944527 | 38035.72 | 0.947159 |
| Romania | 1671.969 | 0.898267 | 2117.189 | 0.992001 | 4450.152 | 0.969168 | 775.4093 | 0.94428 | 3536.105 | 0.968436 | 2908.538 | 0.984939 | 2970.45 | 0.969962 | 1354.464 | 0.985215 | 2184.539 | 0.889246 | 651.4147 | 0.973116 |
| India | 26740.69 | 0.866906 | 26038.41 | 0.90458 | 28120.56 | 0.868558 | 30879.62 | 0.856293 | 30172.63 | 0.880607 | 32096.27 | 0.864628 | 32427.02 | 0.850496 | 30998 | 0.902642 | 32964.09 | 0.903903 | 35305.88 | 0.880268 |
| Egypt | 1454.458 | 0.910107 | 5014.801 | 0.854911 | 3134.967 | 0.89612 | 4475.04 | 0.866355 | 1823.387 | 0.767643 | 6338.392 | 0.928924 | 793.8166 | 0.914896 | 4371.99 | 0.929269 | 3318.488 | 0.91732 | 7847.284 | 0.860965 |
| Saudi Arabia | 5723.754 | 0.96286 | 9389.448 | 0.972382 | 8695.628 | 0.956963 | 9616.636 | 0.923381 | 8540.846 | 0.956426 | 10406.67 | 0.930642 | 9451.943 | 0.957097 | 9818.048 | 0.946917 | 10524.93 | 0.951704 | 7365.579 | 0.984713 |
| Japan | 127.033 | 0.974775 | 1411.46 | 0.950234 | 2357.121 | 0.900003 | 856.8689 | 0.978055 | 1155.087 | 0.977134 | 853.8409 | 0.937967 | 890.435 | 0.948021 | 1314.2 | 0.990891 | 1701.057 | 0.979606 | 758.8102 | 0.936796 |
| Sri Lanka | 3574.6 | 0.959749 | 1946.731 | 0.946785 | 1052.515 | 0.209946 | 1470.191 | 0.00036 | 4432.273 | 0.853986 | 2253.343 | 0.869612 | 2353.647 | 0.812553 | 497.3086 | 0.935267 | 1311.281 | 0.95651 | 106.2703 | 0.768977 |

time-series model without considering the factors that characterise countries or policies taken to find "general rules and features" would mean finding simple rules for each country at a given time. In most simple ways, when the curve is only increasing at the initial spread time.

Last, even if these previous issues are solved, the world is connected, the spatial weights may vary from country to country or day to day based on the restrictions and measures are taken. If there are simple rules that ultimately can fit the entire countries, the challenges would remain in how to weight the changes around the world. Most importantly, one single case in one region could influence the spread elsewhere.

## 5.2 Deployment and online inference

The model is trained on data from different countries, with different measures applied at a different pace. It has also seen data before, during and after lockdown measures of different regions. Moreover, the model has seen data before and after travel restrictions (from January to June) for certain countries. Accordingly, even if measures and restrictions change in the future, this could allow the model to predict future cases by understanding and inferring the change in the measures and their effects in a given country. However, as for the future improvement of the model, more data types, when they are available, could be fed into the model. Besides the policies and restriction that would vary from country to country, real-time mobility data or mobility indices could help the model to forecast new cases in the post-lockdown periods.

Building on how the model could be used for future inference, deploying the model in an online platform is a possible application of this research. This application could assist policy-makers to have a better overview picture of the status of the spread of the virus across the globe to help implement or eliminate a given policy. In Table 4, we show the run-time required for the different tasks on a single GPU. We show that updating the data, computing adjacency, fusing the data, and re-training the model would require less 5 min, whereas utilising the model for inference will require less a second to predict a future step (multi-steps) for all countries.

## 5.3 Limitations and future work

The generative graph of the model along with the other factors has generated good predictions for each country globally (based on trial and errors). However, it remains a challenge that countries with spread over a longer period are more likely to be predicted more accurately than countries with no prior cases, despite how large or small the numbers of cases are. Based on our experiments, the model still understands the pattern in countries that are perceived as outliers, but with lower accuracy. For example in case of Sri Lanka, the strength of the model performance, in term of r-squared, decreases by 23% or the logarithmic error increases by 2.5 folds in comparison to the model performance in predicting spread cases in the United States (see Table 1).

Re-training the model with more data in the future would yield better results at both global and country levels. Besides data improvement, there are three main ways in which the model algorithms can be advanced in future work. First, finding more significant spatial or

**Table 4. Training and inference runtime on a single GPU (Nvidia GTX 2080 Ti).**

| TASK | Runtime |
| --- | --- |
| Data preprocessing and fusion | 32 sec |
| Model training (500 epochs) | 218.94 sec (3.6 min) |
| Model inference | 0.4 sec |
| Model inference and data plotting for each country | 6.3 sec |

demographic factors that show significant associations with the spread may enhance the forecasts of the model. Second, applying the same concept and goals of the model to other subjects of coronavirus could lead to a better understanding of its future. This may include estimating deaths or recovery, bearing in mind the health system capability and capacity, in addition to the governmental responses. Currently, the model is capable of forecasting 10 steps in the future with acceptable accuracy, in which it is validated. With more data on more factors, the introduced method could also lead to better long-term forecast for each country based on the lesson learned from the global and country-level trends.

Last, the method introduced can be used for polices evaluation by changing a given policy for a given country and date to show how the prediction in future is affected. Accordingly, as for future research, exploring the effect of the different urban factors and governmental measures at both global and country levels can be tackled to assist policy-makers to reach optimal measures amid toward reducing the spread of coronavirus. for reaching optimal measures.

## 6. Remarks and lessons learned

In this article, we introduced a novel variational-LSTM autoencoder to predict the spread of coronavirus for different regions/countries across the globe. The introduced learning process and the structure of the data are keys. The model learned from various types of dynamic and static data, including the historical spread data for each country, urban and demographic features such as urban population, population density, and fertility rate, and government responses for each country amid towards mitigating coronavirus outbreak. Also, the model learned to sample different conditions and adjustments of a spatially weighted adjacency matrix among the different infected countries. Overall, the model shows high validation for forecasting the spread at global and country levels, which makes it a useful tool to assist decision and policymaking for the different corners of the globe.

There are several lessons learned while conducting this research. First, concerning urban features, we found several associations of several factors with the spread of coronavirus globally for a specific period of the tested duration. Most significantly, countries with a higher density of population in one $km^2$ and larger portion of the population living in urban areas are associated with higher coronavirus spread with different coefficients, and levels of statistical significance during the examined duration, whereas countries with higher fertility rates are associated with fewer spread cases at the given studied period (22/01/2020-14/06/2020). However, we also found an association with other factors that not used in this research such as migration nets. We found that countries with higher migration flows are associated with higher spread which could also be explained with their likelihood of having a higher influx of job opportunities. Second, concerning the computed adjacency matrix graph, we found that at very short distances among the different infected countries with coronavirus spread, Western European countries (such as Germany, Italy, Spain) are fully or partially connected relative to other countries globally that are same distance they are completely isolated. This can be reflected on the relatively shorter distance–as a physical attribute-as among these countries when it compares to other countries, or the non-physical accessibility of the European market which could lead to a higher influx of migration and accordingly higher spread cases.

## Supporting information

**S1 Data.**
(ZIP)

## Author Contributions

**Conceptualization:** Mohamed R. Ibrahim.

**Data curation:** Mohamed R. Ibrahim.

**Formal analysis:** Mohamed R. Ibrahim.

**Investigation:** Mohamed R. Ibrahim.

**Methodology:** Mohamed R. Ibrahim.

**Project administration:** Mohamed R. Ibrahim.

**Resources:** Mohamed R. Ibrahim.

**Software:** Mohamed R. Ibrahim.

**Supervision:** James Haworth, Aldo Lipani, Tao Cheng, Nicola Christie.

**Validation:** Mohamed R. Ibrahim.

**Visualization:** Mohamed R. Ibrahim.

**Writing – original draft:** Mohamed R. Ibrahim, Nilufer Aslam.

**Writing – review & editing:** Mohamed R. Ibrahim, James Haworth.

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
