## [Decision Letter · Decision Letter 0]

15 Jun 2020

PONE-D-20-10875

Forecasting the spread of coronavirus across the globe with deep learning

PLOS ONE

Dear Dr. Ibrahim,

Thank you for submitting your manuscript to PLOS ONE. After careful consideration, we feel that it has merit but does not fully meet PLOS ONE’s publication criteria as it currently stands. Therefore, we invite you to submit a revised version of the manuscript that addresses the points raised during the review process.

The paper shows a model that is interesting (see comment of reviewer 1). However, the manuscript makes extravagant claims. It also misleads by using linear scale whose results in the most part appear as being close with the real data while in reality are far in a log scale (see comments of reviewer 2). I recommend to rather apply log scales and to claim more realistic discourse closer to the data, emphasizing the limits of the model

We look forward to receiving your revised manuscript.

Kind regards,

Celine Rozenblat

Academic Editor

PLOS ONE

Journal Requirements:

2.We suggest you thoroughly copyedit your manuscript for language usage, spelling, and grammar. If you do not know anyone who can help you do this, you may wish to consider employing a professional scientific editing service.  

3. We note that [Figure(s) 3, 4, 6, 8] in your submission contain [map/satellite] images which may be copyrighted. All PLOS content is published under the Creative Commons Attribution License (CC BY 4.0), which means that the manuscript, images, and Supporting Information files will be freely available online, and any third party is permitted to access, download, copy, distribute, and use these materials in any way, even commercially, with proper attribution. For these reasons, we cannot publish previously copyrighted maps or satellite images created using proprietary data, such as Google software (Google Maps, Street View, and Earth). For more information, see our copyright guidelines: http://journals.plos.org/plosone/s/licenses-and-copyright.

1.    You may seek permission from the original copyright holder of Figure(s) [#] to publish the content specifically under the CC BY 4.0 license.

4. Please ensure that you refer to Figure 4 and 6 in your text as, if accepted, production will need this reference to link the reader to the figure.

Additional Editor Comments (if provided):

The paper shows a model that is interesting (see comment of reviewer 1). However, the manuscript makes extravagant claims. It also misleads by using linear scale whose results in the most part appear as being close with the real data while in reality are far in a log scale (see comments of reviewer 2). I recommend to better apply log scales and to claim more realistic comments closer to the data, emphasizing the limits of the model

Reviewers' comments:

Reviewer's Responses to Questions

**Comments to the Author**

1. Is the manuscript technically sound, and do the data support the conclusions?

Reviewer #1: Yes

Reviewer #2: Partly

2. Has the statistical analysis been performed appropriately and rigorously? 

Reviewer #1: I Don't Know

Reviewer #2: No

3. Have the authors made all data underlying the findings in their manuscript fully available?

Reviewer #1: Yes

Reviewer #2: Yes

4. Is the manuscript presented in an intelligible fashion and written in standard English?

Reviewer #1: Yes

Reviewer #2: Yes

5. Review Comments to the Author

Reviewer #1: The study reports a model that predicts trends of spread of coronavirus at a country level which will be of immense utility value for policymakers and public health planners.

An added utility is that it is based on machine learning and will complement other mathematical and statistical models.

The novelty of the model is that it includes historical data, location, demographic data, and more importantly government measures to mitigate the epidemic (e.g. cancelled public events, travel restrictions, closing of work-places, closure of international travel etc.) and spatial dependence of states to each other. This improves the validity of the model to higher levels of precision. The use of country-specific data has enhanced its relevance and applicability to policymakers at a country level. Its flexibility and ability to refine to the city level is a distinct advantage, because the epidemic often begins in localities and spreads within cities.

The paper is well written and easy to understand. The tables and figures are excellent. The references are up-to-date.

Two points could be explored. How would the model respond to further data on the effectiveness of measures (e.g. travel restrictions as a measure vs. real-time mobility data as an outcome of the measure)? How well would the model fit with countries that are outliers (e.g. Sri Lanka has had less than 1350 cases as of 26th May and 10 deaths. This includes 650+ cases detected within a navy camp and about 300 in Sri Lankan returnees who were working overseas, especially from the Middle East. This demonstrating that their strict policy of curfews coupled with detect, trace contacts, quarantine has essentially limited the spread to clusters with almost zero community spread (https://epid.gov.lk/web/).

Reviewer #2: The manuscript discusses the application of a Long-Short Term Memory autoencoder in the data of Covid-19 with the goal of producing a short (next day) and a longer (10-day) term forecast.

Indeed, the model makes a prediction of the outcome of the coronavirus cases in several countries. However, and given the results shown by the authors themselves, it does not seem to produce the desirable effect and most of the projections seem to be far from acceptable.

My overall comments are the following:

The results are projected in linear figures, while it is obvious that the logarithmic scale is the one which is of interest. Where the results to be transformed in a logarithmic scale (which is of interest for projecting the cases in the next few days) the results from the authors would in many countries show differences from real data. For example, the prediction for Italy would show, in a logarithmic figure that the cases would flatten out soon, while the real data (and the ensuing history which is now known) would have shown the opposite. Countries like Russia, France, the UK and more from the ones mentioned in here, have the same (I would say almost systematic) type of behavior. That is, the model shows almost all of them with a significantly lower estimate of expected cases. Though the actual difference is small, the trend is significantly different. And the trend is more important than the actual number as it is what allows (or enforces) additional measures to be lifted (or taken).

Additionally, the manuscript makes extravagant claims such as the possibility for a 1-3 month prediction by using this method. I feel that the forecasts made in the manuscript are not as good to support such a claim.

On a separate issue, I feel that the reality has taught us that only the initial few cases of a specific country depend on how other countries are going. Upon quarantine measures are taken into effect the evolution of cases in any country depends on the strictness and enforcement of the measures taken. Countries like South Korea and the USA have had (and still have) a totally different history and under no circumstances does the case of one affect the other.

Overall, I feel that the manuscript makes extravagant and largely unsupported by the results and reality claims. It also misleads by using linear scale whose results in the most part appear as being close with the real data while in reality are far in a log scale.

As such, I cannot recommend publication of this manuscript.

6. PLOS authors have the option to publish the peer review history of their article (what does this mean?). If published, this will include your full peer review and any attached files.

Reviewer #1: No

Reviewer #2: No

---

## [Author Response · Author response to Decision Letter 0]

23 Jun 2020

Dear Editor and reviewers, 

Thanks for your comments and feedback. We have modified several aspects in the article:

1- We have updated the data to June, to make the article up-to-date.

2- We have applied a logarithmic scale for the outputted data alongside the actual scale (linear scale). We believe that both have very different applications. Understanding the pattern is one thing while giving the ability to predict actual values is also vital for many applications. 

On a side note, a logarithmic function tends to smooth out any given function, by reducing the data dimensionality and removing outliners. This makes it simpler to forecast than the actual number, unlike what reviewer 2 mentioned (In fact, a log output would tend to show a better performance of our model than the actual data). Please, see the added figures. The log functions tend to widen the difference of the earlier points of low values, whereas close the different at the end of the curve. This shows how the model learns the pattern of the data after the initial zero cases of most countries (after having enough cases to extract patterns).

3- Besides our evaluation metrics, we have added also a Mean squared logarithmic error (MSLE) to compare the results of different countries. It is a relative error, which could be a good indicator for comparing the performance of the model in different countries despite how large or small the values of the cases in a country. 

4- We have added two new discussion points: to discuss outliners, and the online inference of the model and its flexibility to different data sources as suggested by reviewers 1

All figures including maps are within our copyrights, we did not use google maps or any copyrighted software to visualise them. We created them using open-access python libraries- Basemap and Matplotlib. 

Please, see a detailed response to each comment below.

Additional Editor Comments:

The paper shows a model that is interesting (see comment of reviewer 1). However, the manuscript makes extravagant claims. It also misleads by using linear scale whose results in the most part appear as being close with the real data while in reality are far in a log scale (see comments of reviewer 2). I recommend to better apply log scales and to claim more realistic comments closer to the data, emphasizing the limits of the model

We have applied a logarithmic scale for the outputted data alongside the actual scale (linear scale). We believe that both have very different applications. Understanding the pattern is one thing while giving the ability to predict actual values is also vital for many other applications that require a short-term forecast of accurate estimation per day.

We also discussed more points to emphasise the limits of the model in the discussion section.

Reviewer #1: 

The study reports a model that predicts trends of the spread of coronavirus at a country level which will be of immense utility value for policymakers and public health planners.

An added utility is that it is based on machine learning and will complement other mathematical and statistical models.

The novelty of the model is that it includes historical data, location, demographic data, and more importantly government measures to mitigate the epidemic (e.g. cancelled public events, travel restrictions, closing of work-places, closure of international travel etc.) and spatial dependence of states to each other. This improves the validity of the model to higher levels of precision. The use of country-specific data has enhanced its relevance and applicability to policymakers at a country level. Its flexibility and ability to refine to the city level is a distinct advantage, because the epidemic often begins in localities and spreads within cities.

The paper is well written and easy to understand. The tables and figures are excellent. The references are up-to-date.

Two points could be explored. How would the model respond to further data on the effectiveness of measures (e.g. travel restrictions as a measure vs. real-time mobility data as an outcome of the measure)? 

This is an interesting and valid point for discussion. The model is trained on data from different countries, with different measures applied at a different pace. It has also seen data before, during and after lockdown measures of different regions. Moreover, the model has seen data before travel restriction and after travel restriction (from January to June) for certain countries. Accordingly, even if measures and restrictions change in future, this could allow the model to predict future cases by understanding and inferring the change in the measures and their effects in a given country. However, as for the future improvement of the model, more data types -when they are available- could be feed for the model. Besides the policies and restriction that would vary from country to country, real-time mobility data or mobility indices could help the model to forecast new cases in the post-lockdown periods. 

Building on how the model could be used for future inference, deploying the model in an online platform is a possible application of this research. This application could assist policy-makers to have a better overview picture of the status of the spread of the virus across the globe, nevertheless, implement or eliminate a given policy. In table 3, we show the run-time required for the different tasks on a single GPU. We show that updating the data, computing adjacency, fusing the data, and re-training the model would require less 5 min, whereas utilising the model in the inference model will require less a second to predict a future step (multi-steps) for all countries. 

We have added a discussion point in the discussion section, addressing how the model can be used for online inference in a data stream. 

How well would the model fit with countries that are outliers (e.g. Sri Lanka has had less than 1350 cases as of 26th May and 10 deaths. This includes 650+ cases detected within a navy camp and about 300 in Sri Lankan returnees who were working overseas, especially from the Middle East. This demonstrating that their strict policy of curfews coupled with detect, trace contacts, quarantine have essentially limited the spread to clusters with almost zero community spread (https://epid.gov.lk/web/).

This seems to be an interesting case. We have discussed the limitation of the model to outliers, in the discussion section. We also added the case of Sri Lanka in the results of the single-step and multi-step models to show the performance of the model and how the model performs in countries with fewer cases.

Overall, the model would perform less accurately in countries with fewer data points for a given country (countries with a spread that started at a very late stage). However, the model would not be affected by how high or low the values per day as long as there are enough data points for a country to learn its pattern. 

Based on our experiments, the model still understands the pattern in countries that are perceived as outliers, but with lower accuracy. For example in case of Sri Lanka, the strength of the model performance, in term of r-squared, decreases by 23% percent or the logarithmic error increases by 2.5 folds in comparison to the model performance in predicting spread cases in the United States (See table 1). We further discussed this issue in 5.3 Limitation and future work

Reviewer #2: 

The results are projected in linear figures, while it is obvious that the logarithmic scale is the one which is of interest. Where the results to be transformed in a logarithmic scale (which is of interest for projecting the cases in the next few days) the results from the authors would in many countries show differences from real data. For example, the prediction for Italy would show, in a logarithmic figure that the cases would flatten out soon, while the real data (and the ensuing history which is now known) would have shown the opposite. Countries like Russia, France, the UK and more from the ones mentioned in here, have the same (I would say almost systematic) type of behavior. That is, the model shows almost all of them with a significantly lower estimate of expected cases. Though the actual difference is small, the trend is significantly different. And the trend is more important than the actual number as it is what allows (or enforces) additional measures to be lifted (or taken).

Given that understanding the pattern is important as well as the true values, we have plotted the results in true figures and logarithmic scale. It is up to the readers- or policy-makers -to apply the model in whichever form that suit the application. 

Concerning the trend more important than the actual numbers: Understanding the trend is one application of a predictive model and could be important for drawing policies or remove certain regulation, which we have provided in the new figures for each country which the model shows a good understanding of the pattern. Since that log function tends to reduce the dimensionality of data, many countries will follow the same trend, even if they have an extremely different count of cases. This also can be seen as misleading information. For instance, if you trying to estimate how many bed or ventilations in the hospitals that are needed in the next days, then the actual number per day is more vital than the pattern. Even if the log scale is transformable to a normal scale, the small loss at a log scale- if the model is trained in it- would lead to a large loss in normal scale. Nevertheless, the latter (Estimating numbers in normal scale) is missing in the literature as most of the model would rather project a pattern, than provide a predictive model that able to predict day by day.

Additionally, the manuscript makes extravagant claims such as the possibility for a 1-3 month prediction by using this method. I feel that the forecasts made in the manuscript are not as good to support such a claim.

We mentioned in the early manuscript in future work section: “Put all together, more data, more factors, different forecasting models could also lead to better long-term forecast (1-3 months) for each country based on the lesson learned from the global and country-level trends of spread.”

We have re-stated our statements to make sure that the scope and the limitation are clear, and how this could be reached in the future using our model. 

Concerning the statement: “I feel the forecasts made in the manuscript are not as good”, we have provided several empirical methods to quantify the losses and the accuracies of our model, in the addition to the descriptive analysis of the plotted figures and maps which shows with more data in the future- similar to any machine learning models- the model would be capable of forecasting the long term pattern if fed with more data on the upcoming months, which supports our statement of the long-term forecasts.

On a separate issue, I feel that the reality has taught us that only the initial few cases of a specific country depend on how other countries are going. 

We are not sure if we fully understood this statement. Initial cases could be a good indicator of how a given country can lead to- for understanding other countries. However, what happens when a country that has similar initial cases to another country, took different measures? We don’t think initial cases is enough to predict the complexity of reality as we mention in the articles each country has unique factors, measures, spread patterns, and adjacency link to other countries. Only, the combinations of this, or even more factors, could lead to better prediction. 

We have clarified that in the new submission. 

Upon quarantine measures are taken into effect the evolution of cases in any country depends on the strictness and enforcement of the measures taken. 

We agree that strictness of the measure during the quarantine time could be an indicator however, it is never be quantified or supported by empirical methods. Nevertheless, there are other factors, even with strict enforcement that could lead to the evolution of cases in a given country. For instance, the effect of going to grocery stores is not yet measured, whereas it takes place in every country despite the enforcement of the quarantine measures and could lead to exposures. Second, the number of testing per day -despite how strict the enforcement of measures could lead to adding more numbers of cases in a given country, despite how strict the measure is. Third, people moving by car or bus to adjust countries, which we think it is the case in the EU countries, despite how strict the measures countries are taking. 

Based on our empirical findings, understanding adjacency of countries has improved the prediction of the model – in comparison to without the adjacency of countries- and we have mentioned that in our method.

Countries like South Korea and the USA have had (and still have) a totally different history and under no circumstances does the case of one affect the other.

We agree that countries have different patterns due to their demographics features, urban features, spread historical data, and applied policies and measures. We never mentioned anything regarding South Korea and the USA having a similar history. However, one case elsewhere is enough to cause spread in another country by mobility despite the different nature of each county –the reality has taught us that. Understanding the adjacency between countries means including factors due to mobility that is not necessarily based on travel by flights. It could, for instance, travel by car (which could be the case across Europe, or across the US states), even during the US taking an international strict measures, but moving around the country or adjacent countries was not yet limited. Even if cases are few, they remain as important risk factors that could be neglected.

---

## [Decision Letter · Decision Letter 1]

22 Sep 2020

PONE-D-20-10875R1

Variational-LSTM Autoencoder to forecast the spread of coronavirus across the globe

PLOS ONE

Dear Dr. Ibrahim,

Thank you for submitting your manuscript to PLOS ONE. After careful consideration, we feel that it has merit but does not fully meet PLOS ONE’s publication criteria as it currently stands. Therefore, we invite you to submit a revised version of the manuscript that addresses the points raised during the review process.

Thanks for having improved the explanations and output of the model. It remains some minor remarks from the 2 reviewers.

Reviewer 1: asks to mention other possible variables (maybe at the end of the paper in conclusion to explain why the model fitted lower with some cases and give future possible improvements)

Reveiwer 2: be more modest on the result and improve the Log graphs by putting "real values": Also I would add that log graphs must have the background of lines with uneven distances, which shows  obviously at the first glance that it is log (do you see what I mean? If it is not clear I can send you an example of these kinds of graphs by email: please contact me directly)

Please see below the details of the feed-back of the 2 reviewers. I hope that you will be able to address these last remarks quickly

We look forward to receiving your revised manuscript.

Kind regards,

Celine Rozenblat

Academic Editor

PLOS ONE

Additional Editor Comments (if provided):

Thanks for having improved the explanations and output of the model. It remains some minor remarks from the 2 reviewers.

Reviewer 1: asks to mention other possible variables (maybe at the end of the paper in conclusion to explain why the model fitted lower with some cases and give future possible improvements)

Reveiwer 2: be more modest on the result and improve the Log graphs by putting "real values": Also I would add that log graphs must have the background of lines with uneven distances, which shows obviously at the first glance that it is log (do you see what I mean? If it is not clear I can send you an example of these kinds of graphs by email: please contact me directly)

Please see below the details of the feed-back of the 2 reviewers. I hope that you will be able to address these last remarks quickly

Reviewers' comments:

Reviewer's Responses to Questions

**Comments to the Author**

1. If the authors have adequately addressed your comments raised in a previous round of review and you feel that this manuscript is now acceptable for publication, you may indicate that here to bypass the “Comments to the Author” section, enter your conflict of interest statement in the “Confidential to Editor” section, and submit your "Accept" recommendation.

Reviewer #1: All comments have been addressed

Reviewer #2: (No Response)

2. Is the manuscript technically sound, and do the data support the conclusions?

Reviewer #1: Yes

Reviewer #2: Partly

3. Has the statistical analysis been performed appropriately and rigorously? 

Reviewer #1: I Don't Know

Reviewer #2: No

4. Have the authors made all data underlying the findings in their manuscript fully available?

Reviewer #1: Yes

Reviewer #2: Yes

5. Is the manuscript presented in an intelligible fashion and written in standard English?

Reviewer #1: Yes

Reviewer #2: Yes

6. Review Comments to the Author

Reviewer #1: The authors have responded to the point raised by the reviewers. The paper clearly states the aims, the methods used, the results obtained and discusses the results while indicating the drawbacks of the study. As regards to urban features, the authors mention the presence of "...a wide range of factors, however, we only selected three factors; 1) Population density, 2) fertility rate and 3) urban population". It would be good to mention some of the other urban features that were ignored due to lack of a correlation etc.

Reviewer #2: I believe that the authors have taken into account my comments and suggestions, albeit, not all of them exactly as they were intended.

Specifically, my point was to have the presentation of the results in a log-normal fashion. The trend up to then would have been misleading to a reader in some cases (e.g. Russia, France, Romania, and Italy with the previous data). Thus, I had only expected the authors to show log-normalized versions of their linear analysis results. The log normal representation would emphasize on the points where there is a change in the trend. E.g. in countries where you were far from the logarithmic "plateau" (Russia) you could foresee that the situation was not coming under control anytime soon in a log normal representation, while in your forecasts this was not obvious. A log normal representation of your RMSE results would have shown that you were unjustly predicting a "plateau". Also, in general, the trend changes in the beginning of all actual country cases real data could not be seen in linear scales, while in log normal trend changes were clearly shown.

The log-normal analysis with the new MSLE presented in the revised version, gets results that seem much more confident in the end of their set. However, the results show very bad confidence in the beginning. By using more recent data, most countries now exhibit smaller changes in the curvature of their cases lines, especially given the much bigger training set usable for the method. Additionally, indeed, as the authors point out the logarithmic value of a number suppresses the dimensionality of the results. The effort to "flatten the curve" though by many countries wanting to control Covid-19, was meant for the log-normal cases lines (same amount of new cases per day) and not for the linear ones (which would mean no cases at all per day).

I feel that the method presented is not very accurate in following abrupt changes in the curvature of a cases line (my point to begin with). This comment can be made even for the linear analysis (RMSE), where changes are followed always with a delay by the forecast. This, in my opinion, reduces the strength of the analysis greatly. It essentially translates to an inability to forecast reliably in any point in time, other than that where only linear or almost linear changes occur. Forecasting only the linear part of a line, or very smoothly changing lines, is not a major achievement and this part of the manuscripts' analysis is not very appealing in my opinion.

Despite this, and given that a comment is included that the method fails to predict abrupt changes (even more for small datasets), the analysis can indeed present valid results for all other cases, and the technical standards used are high enough for it to warrant publication in PLOS. It is simply not ground breaking research in my opinion.

My only other suggestion now is that the log normal figures of the manuscript should be presented in their y-axis as is standardized in such cases (1,10,100,1000, 10K, 100K, etc), and not in their logarithmic results (now used 0, 1, 2, 3, 4, 5 respectfully).

I urge the authors to check what has been done for example for Covid-19 in the worldometers website. They can perhaps even see how different the log-normal (mentioned as logarithmic in there) results are than the linear ones (especially early on for each country), and how the log-normal representation pointed to moments where trends changed significantly.

7. PLOS authors have the option to publish the peer review history of their article (what does this mean?). If published, this will include your full peer review and any attached files.

Reviewer #1: No

Reviewer #2: No

---

## [Author Response · Author response to Decision Letter 1]

28 Sep 2020

Dear Editor and reviewers, 

Thanks for your comments and feedback. We have addressed the two comments raised by the reviewers. 

Reviewer 1: asks to mention other possible variables (maybe at the end of the paper in conclusion to explain why the model fitted lower with some cases and give future possible improvements)

We have added discussed the variables that we found insignificant in section 3.1.2 Urban features data.

Reviewer 2: be more modest on the result and improve the Log graphs by putting "real values": Also I would add that log graphs must have the background of lines with uneven distances, which shows obviously at the first glance that it is log (do you see what I mean? If it is not clear I can send you an example of these kinds of graphs by email: please contact me directly)

We have modified the interpretation of the result to be more modest by removing unnecessary words. We have changed all the logarithmic graphs to be with an unevenly distributed y-axis.

Regards,

The authors.

---

## [Decision Letter · Decision Letter 2]

26 Nov 2020

PONE-D-20-10875R2

Variational-LSTM Autoencoder to forecast the spread of coronavirus across the globe

PLOS ONE

Dear Dr. Ibrahim,

Thank you for submitting your manuscript to PLOS ONE. After careful consideration, we feel that it has merit but does not fully meet PLOS ONE’s publication criteria as it currently stands. Therefore, we invite you to submit a revised version of the manuscript that addresses the points raised during the review process.

Thanks for having address the previous questions. Please  address the final requests of reviewer 3 that are very relevant

We look forward to receiving your revised manuscript.

Kind regards,

Celine Rozenblat

Academic Editor

PLOS ONE

Reviewers' comments:

Reviewer's Responses to Questions

**Comments to the Author**

1. If the authors have adequately addressed your comments raised in a previous round of review and you feel that this manuscript is now acceptable for publication, you may indicate that here to bypass the “Comments to the Author” section, enter your conflict of interest statement in the “Confidential to Editor” section, and submit your "Accept" recommendation.

Reviewer #2: All comments have been addressed

Reviewer #3: (No Response)

2. Is the manuscript technically sound, and do the data support the conclusions?

Reviewer #2: Yes

Reviewer #3: Partly

3. Has the statistical analysis been performed appropriately and rigorously? 

Reviewer #2: I Don't Know

Reviewer #3: Yes

4. Have the authors made all data underlying the findings in their manuscript fully available?

Reviewer #2: Yes

Reviewer #3: Yes

5. Is the manuscript presented in an intelligible fashion and written in standard English?

Reviewer #2: Yes

Reviewer #3: Yes

6. Review Comments to the Author

Reviewer #2: I believe that all of my previous comments have been addressed and thus I feel that the manuscript can be published in the journal.

Reviewer #3: The paper presents a model to predict coronavirus spread based on machine learning and it could be another important tool in the hands of policymakers to better attempt to control the epidemic spread.

The model uses historical data, location, demographic data to feed the model and more importantly it uses government measures to control the epidemic and interdependency of the countries/cities to address the influence of each in the state of the epidemic of the others.

The paper is well written and is easy to understand and to the best of my knowledge is a good contribution to the COVID research landscape.

However I think some points should be more highly discussed.

The authors claim that the use of demographic data is one of the advantages of the proposed method. In section 3.1.2 the authors present the factors from the demographic data they included in the model and how it is correlated with the covid cases spread over time, but afterwards no result is presented confirming the impact of these factors in the prediction. I have the intuition that some if not all of these factors of demographic data are somehow present in the COVID-19 confirmed cases data.

Following the same line of thinking, the interaction between countries, that I think is one of the most important considerations of the paper, does not explore its impact on the results of the model. I think some comparative results may be included considering different adjacency matrices, like the ones presented in section 3.1.4, and the absence of it.

Also regarding interaction between countries and adjacency matrices, in some early papers of COVID-19, for instance the ones presented by Vittoria Colizza on risk of importation (https://www.thelancet.com/journals/lancet/article/PIIS0140-6736(20)30411-6/fulltext, https://journals.plos.org/plosmedicine/article?id=10.1371/journal.pmed.1003193) is used the data of air travel as a proxy of the interaction. Why is this not considered in this model?

Another important advantage of the model is the inclusion of governmental mitigation measures. In the paper (https://www.thelancet.com/journals/laninf/article/PIIS1473-3099(20)30785-4/fulltext?fbclid=IwAR3tcQgOb7CUa3bAxgOL7bsqRPj5jJn-UZI4KcpzTBfIEgt98YWucuZU2Zg) it is shown that most measures have an impact on the reproduction number, and therefore in the cases after 7-14 days. Considering that the predictions of the model are somehow validated to be good till 10 days in advance there is any result that supports that the prediction is being influenced by the measures?. Could it quantify the change in the behavior of the prediction after a governmental measure is taken?.

7. PLOS authors have the option to publish the peer review history of their article (what does this mean?). If published, this will include your full peer review and any attached files.

Reviewer #2: No

Reviewer #3: No

---

## [Author Response · Author response to Decision Letter 2]

2 Dec 2020

Dear editor and reviewers,

We would like to thank you for your time and effort to review this article for the third round. 

We added a new table (Table 1) to explore the impact of the adjacency matrices and variational autoencoder components on the global results of the model as suggested by Reviewer 3. 

Reviewer 3 has suggested different discussion points with reference to several articles that have been published in July or even last month (October), whereas this presented article was submitted to the journal in April. Given the length of time our article has been under review and the unprecedented pace of progress in this field, we believe it is unfair to require us to substantially modify the paper to account for new literature at the third review round.

The article has been through three different minor revisions since the first submission and all the raised comments were minor comments from different reviewers that focus on addressing viewpoints and discussion. While we believe all of the raised comments are valuable and valid discussion points, they do not contribute to the advancement of the introduced method which is our main contribution in this article. 

Please find our detailed responses and, where we feel it is necessary, rebuttals for the raised discussion points below. We would appreciate it if the journal makes a final decision concerning the presented article. 

Regards,

Authors

Reviewer #3: The paper presents a model to predict coronavirus spread based on machine learning and it could be another important tool in the hands of policymakers to better attempt to control the epidemic spread.

The model uses historical data, location, demographic data to feed the model and more importantly it uses government measures to control the epidemic and interdependency of the countries/cities to address the influence of each in the state of the epidemic of the others.

The paper is well written and is easy to understand and to the best of my knowledge is a good contribution to the COVID research landscape.

However, I think some points should be more highly discussed.

The authors claim that the use of demographic data is one of the advantages of the proposed method. In section 3.1.2 the authors present the factors from the demographic data they included in the model and how it is correlated with the covid cases spread over time, but afterwards no result is presented confirming the impact of these factors in the prediction. I have the intuition that some if not all of these factors of demographic data are somehow present in the COVID-19 confirmed cases data.

It has been mentioned in the article (Section 3.1.2) that there are two main reasons for selecting these factors: 1) their correlation with the spread, and 2) their enhancement of the model outcomes based on trial and error. The impact of the factors has been seen during the creation of the model and how the model becomes more reliable. In other words, these factors provide a unique set of features for the different countries that the model self-learn from it through the introduced algorithm. This also has been mentioned in Model Hypothesis, see section 2.1 Hypothesis and assumptions.

Following the same line of thinking, the interaction between countries, that I think is one of the most important considerations of the paper, does not explore its impact on the results of the model. I think some comparative results may be included considering different adjacency matrices, like the ones presented in section 3.1.4, and the absence of it.

We have added a new table to compare the results with and without the introduced adjacency matrix at a global scale. The results of the introduced method with the variational autoencoder component and the introduced adjacency matrix reduces the losses of the same model without these components with an RMSE loss value of 174.3 (See table 1). Also, for further analysis, the below figure shows the outcomes of the model without adjacency similar to the one introduced in the article with adjacency matrix and VAE (Figure 9). It also confirms how the results improved by applying the adjacency matrix. 

Also regarding interaction between countries and adjacency matrices, in some early papers of COVID-19, for instance, the ones presented by Vittoria Colizza on the risk of importation (https://www.thelancet.com/journals/lancet/article/PIIS0140-6736(20)30411-6/fulltext, https://journals.plos.org/plosmedicine/article?id=10.1371/journal.pmed.1003193) is used the data of air travel as a proxy of the interaction. Why is this not considered in this model?

At the time of writing the article, it was not possible to obtain open access flight data at the global level in a timely fashion. Moreover, there are many more transportation modes that may be used for international travel at the continental level that would not be captured in the flight data. Attempting to collect this data on the global scale would sacrifice the timeliness of our research in a rapidly evolving situation. Therefore, we developed our method to infer mobility from the data. The purpose of using a variational autoencoder in this model along with the adjacency matrix (as explained in the paper) is to generate different weights between the countries that represent the mobility flow, not only air travel but all types of mobility. The data taught us that even when there were air travel restrictions, cases remain increasing in certain countries, for instance, among Western European countries, given that distances are shorter and people can travel with cars and coaches. Cargo flights were still active, etc. Accordingly, using air travel alone as a proxy could be a limitation as well as an advantage.

It worth mentioning that this article has been submitted to the journal in April, whereas the second article you refer to was published on July 17, 2020. Given the rapid pace of developments in this field, we feel it is a little unfair to be asked to consider new literature at the third round of review that were not mentioned in previous rounds. 

Another important advantage of the model is the inclusion of governmental mitigation measures. In the paper (https://www.thelancet.com/journals/laninf/article/PIIS1473-3099(20)30785-4/fulltext?fbclid=IwAR3tcQgOb7CUa3bAxgOL7bsqRPj5jJn-UZI4KcpzTBfIEgt98YWucuZU2Zg) it is shown that most measures have an impact on the reproduction number, and therefore in the cases after 7-14 days.

Again, the suggested article is published on October 22, 2020, whereas our paper has been submitted to the journal in April. The findings of this research were not available when the article was submitted or after previous rounds of review, nor affect the introduced method presented in this article.

 Considering that the predictions of the model are somehow validated to be good till 10 days in advance there is any result that supports that the prediction is being influenced by the measures?. Could it quantify the change in the behaviour of the prediction after a governmental measure is taken?.

We think this is a good idea which we have discussed in future works. The introduced method focuses on predicting the spread cases globally, as a novel method to predict spread cases bearing in mind urban factors, governmental policies, and the adjacencies among different countries with deep learning. Quantifying the change in the behaviour and measuring the effect of governmental policies is a field of research in its own right, which it beyond the scope of the presented article (See section 5.3 Limitations and future work).

---

## [Decision Letter · Decision Letter 3]

14 Jan 2021

Variational-LSTM Autoencoder to forecast the spread of coronavirus across the globe

PONE-D-20-10875R3

Dear Dr. Ibrahim,

We’re pleased to inform you that your manuscript has been judged scientifically suitable for publication and will be formally accepted for publication once it meets all outstanding technical requirements.

Kind regards,

Celine Rozenblat

Academic Editor

PLOS ONE

Additional Editor Comments (optional):

Reviewers' comments:

Reviewer's Responses to Questions

**Comments to the Author**

1. If the authors have adequately addressed your comments raised in a previous round of review and you feel that this manuscript is now acceptable for publication, you may indicate that here to bypass the “Comments to the Author” section, enter your conflict of interest statement in the “Confidential to Editor” section, and submit your "Accept" recommendation.

Reviewer #2: All comments have been addressed

2. Is the manuscript technically sound, and do the data support the conclusions?

Reviewer #2: Yes

3. Has the statistical analysis been performed appropriately and rigorously? 

Reviewer #2: Yes

4. Have the authors made all data underlying the findings in their manuscript fully available?

Reviewer #2: Yes

5. Is the manuscript presented in an intelligible fashion and written in standard English?

Reviewer #2: Yes

6. Review Comments to the Author

Reviewer #2: All of my previous comments have been addressed. I feel that the results presented will be of use to members of the community and, thus, support the publication of this paper as is.

7. PLOS authors have the option to publish the peer review history of their article (what does this mean?). If published, this will include your full peer review and any attached files.

Reviewer #2: No

---

## [Editor Report · Acceptance letter]

18 Jan 2021

PONE-D-20-10875R3 

Variational-LSTM Autoencoder to forecast the spread of coronavirus across the globe 

Dear Dr. Ibrahim:

I'm pleased to inform you that your manuscript has been deemed suitable for publication in PLOS ONE. Congratulations! Your manuscript is now with our production department. 

Kind regards, 

on behalf of

Prof. Celine Rozenblat 

Academic Editor

PLOS ONE